# A Micromechanical Study of Interactions of Cyanate Ester Monomer with Graphene or Boron Nitride Monolayer

**DOI:** 10.3390/ma17010108

**Published:** 2023-12-25

**Authors:** Geeta Sachdeva, Álvaro Lobato, Ravindra Pandey, Gregory M. Odegard

**Affiliations:** 1Department of Physics, Michigan Technological University, Houghton, MI 49931, USA; 2MALTA-Consolider Team and Departamento de Química Física y Analítica, Universidad de Oviedo, 33006 Oviedo, Spain; 3Department of Mechanical Engineering and Engineering Mechanics, Michigan Technological University, Houghton, MI 49931, USA

**Keywords:** cyanate-ester, graphene, pull-apart, elastic stiffness, density functional theory

## Abstract

Polymer composites, hailed for their ultra-strength and lightweight attributes, stand out as promising materials for the upcoming era of space vehicles. The selection of the polymer matrix plays a pivotal role in material design, given its significant impact on bulk-level properties through the reinforcement/polymer interface. To aid in the systematic design of such composite systems, molecular-level calculations are employed to establish the relationship between interfacial characteristics and mechanical response, specifically stiffness. This study focuses on the interaction of fluorinated and non-fluorinated cyanate ester monomers with graphene or a BN monolayer, representing non-polymerized ester composites. Utilizing micromechanics and the density functional theory method to analyze interaction energy, charge density, and stiffness, our findings reveal that the fluorinated cyanate-ester monomer demonstrates lower interaction energy, reduced pull-apart force, and a higher separation point compared to the non-fluorinated counterpart. This behavior is attributed to the steric hindrance caused by fluorine atoms. Furthermore, the BN monolayer exhibits enhanced transverse stiffness due to increased interfacial strength, stemming from the polar nature of B–N bonds on the surface, as opposed to the C-C bonds of graphene. These molecular-level results are intended to inform the design of next-generation composites incorporating cyanate esters, specifically for structural applications.

## 1. Introduction

Deep space exploration into the solar system requires technology advancements for space vehicles that can sustain a voyage with minimal mass and volume. Polymer matrix composites, especially those reinforced with carbon fibers, have gained popularity in aerospace applications in recent years due to their high stiffness-to-weight ratio, which results in significant weight and fuel savings on commercial and military aircraft [1,2,3]. It has been recognized that existing carbon fiber-based composite materials are insufficient for a manned mission to Mars [4]. However, one expects nanomaterials, including graphene and carbon nanotubes (CNTs), can exceed the mechanical properties of carbon fibers used for structural reinforcements in composites [5]. Essentially, composites are engineered materials that consist of at least two materials that are significantly different in terms of their chemical or physical properties.

Because of their excellent mechanical properties, CNTs are a good reinforcement for high-performance polymer matrix composites. With a Young’s modulus of approximately 1 TPa, carbon nanotubes serve as effective reinforcements for materials with exceptional stiffness and strength. However, CNTs dispersed in a polymer matrix have lower stiffness and strength than isolated ones [6]. The challenge related to the dispersion of CNTs arises from inadequate noncovalent bonding between neighboring CNTs, resulting in the deformation-induced movement of CNTs [7]. On the other hand, CNTs with a large diameter (~10 nm), also called flattened CNTs (flCNTs), observed in nanocomposites formed with highly aligned thin CNT films and bismaleimide (BMI) (a type of resin) have been found to have remarkably high tensile strength compared to that of small-diameter CNTs because of their large contact area, and hence a higher degree of noncovalent type of bonding among themselves [6]. For example, the mechanically stretched sheets of flCNTs combined with BMI resin were 78% and 283% greater in Young’s modulus and tensile strength, respectively, than sheets of randomly oriented CNT with BMI resin. These flCNT bundles can thus serve as highly effective reinforcing materials [6].

FlCNTs are stacked layers of two-dimensional graphene [8] in which van der Waals forces dominate the interlayer interactions. This weak attractive interaction compensates for the energy loss from forming reactive edges and provides stability to flCNTs [9]. In the present study, we show that graphene representing the graphitic structure of flCNTs can provide excellent interfacial strength in forming polymer composites.

The question now becomes “Which polymer matrix is most compatible with graphene?” The efficacy of load transfer between composite constituents is determined by the binding strength (or interaction strength at the interface). In this paper, we aim to gain a comprehensive understanding of the fundamental interactions occurring at the monomer–monolayer level during the reinforcement process. By narrowing our focus to the monomer stage, we can elucidate the molecular-level mechanisms, surface energetics, and bonding characteristics that play a major role in the overall performance of the cured polymer composites.

Previously, Sachdeva et al. [10] used density functional theory (DFT) to investigate the interaction properties and mechanical characteristics of two different (epoxy and BMI) monomers interacting with graphene. They demonstrated that various functional groups (e.g., R-O-R in epoxy resin) significantly affect the interfacial interaction energy, pull-apart force, and separation point. Although the reported results are essential in determining the behavior of monomer/graphene interfaces, it is unclear how it interacts with other monomer systems that could be employed in graphene-based polymer composites.

In general, it is expected that functional groups containing nitrogen will be most compatible with graphene-based composites due to the interaction between amide/amine and the orbitals of graphitic structures (i.e., NH_2_–π interaction) [11]. Accordingly, aromatic cyanate esters, Primaset PT-30 and AroCy F-10, consisting of nitrogen-containing functional groups, can be candidate matrices for such composites. These ester resins have evolved as a distinct and new category of thermosetting resins with excellent performance employed in structural applications as matrices [12]. The cyanate ester resins have some basic features, such as low moisture absorption, good electrical qualities, easy processability, good flammability characteristics, high service temperature, and toughness, which makes them ideal composite matrices and a competitor for epoxy and bismaleimides resins in structural applications [13,14,15,16].

Cyanate esters are characterized by a phenol backbone with cyanate (-OCN) groups attached at each end of the monomer. AroCy F-10 is a fluorinated cyanate ester (Figure 1a) with a tensile strength of 75 MPa, an elastic modulus of 3.11 GPa, and a maximum strain of 2.8%, and fracture toughness of 140 Jm^−2^ [17]. Primaset PT is a non-fluorinated cyanate ester (Figure 1b) with a tensile strength of 77 MPa, a tensile modulus of 4.0 GPa, a strain-to-break of 2.0%, and a flexural strength of 112 MPa [18]. In this work, we consider the complexes consisting of an ester monomer and graphene (or BN) monolayer to provide an atomistic description of the interface via interaction energy, the density of states, and Bader’s charge [19] and the population of atoms [10]. Subsequently, the mechanical response of these esters and graphene (or BN) monolayer complexes in terms of separation point, transverse strength, and stiffness are characterized using pull-apart simulations [10]. It is again noteworthy that we intend to focus on the individual monomer interaction at the reinforcement (monolayer) surface rather than the cured polymer composites in the present study.

Since boron nitride (BN) has a graphene-like structure with regularly stacked planar networks of hexagons [20,21], its interactions with the ester monomers were also investigated. It exhibits a polar bonding character, which is likely to have a distinct nature of interaction with various monomers than graphene.

## 2. Computational Details

The Vienna Ab-initio Simulation Package (VASP) [22] was used to perform DFT calculations utilizing projector augmented wave (PAW) [23] potentials. The Perdew, Burke, and Ernzerhof (PBE) parameterization of the generalized gradient approximation (GGA) was employed for the exchange-correlation functional [24], and van der Waals interactions were incorporated using Grimme’s D2 technique [25]. Additionally, we used a Γ-centered k-point grid with a plane-wave cutoff of 500 eV and convergence criterion of 10^−5^ eV for energy. The conjugate gradient (CG) algorithm was used to fully relax local energy-minimum structures with atoms with forces smaller than 0.001 eV/Å.

We employed a 30 Å × 30 Å × 20 Å supercell and applied the periodic boundary conditions to mimic the system as a non-polymerized composite system. To ensure that periodic images do not interfere with each other, a vacuum of 20 Å has been applied along the *z*-axis. The monomer is placed on top of the monolayer at a distance of ~2.5 Å, i.e., slightly higher than the nearest monomer and monolayer atoms van der Waal radii [26]. Then, the geometry optimization was performed to find the equilibrium configuration of the ester/graphene (or BN monolayer) complex.

The commonly used “pull-apart” experimental setup was utilized to determine the complex’s mechanical response [27,28,29,30]. The pull-apart simulation setup schematic to derive the force–strain relationship is shown in Appendix A. At first, starting from an optimized geometry, the monomer moves away perpendicularly to the monolayer surface with a step size of ≈0.02 Å. The calculated variation in energy as a function of each incremental step referred to as (out-of-the-plane) transverse strain is shown in Appendix A. This is how the transverse strain, *ε*, is defined as *ε* = (*l* − *l*_0_)/*l*_0_, where *l* represents the distance of the monomer from the surface for a given step, and *l*_0_ represents the distance in the equilibrium configuration for the complex. It is noteworthy that the equilibrium distance *l*_0_ is the distance between the nearest atoms of the monomer and the monolayer. Also, the associated change in the energy is defined as *E*s = *E*(*ε*) − *E*(0), with *E*(0) being the energy related to the equilibrium configuration. Note that the configuration optimization of the monomer is not performed at each step as it moves away from the surface. The strain-energy data is then obtained by moving one component (monomer) of the complex in one dimension (*z*-axis).

A force–strain curve, derived from the derivative of the strain-energy curve, is used to calculate the mechanical response of the complex. The one-dimensional spinodal equation of state (1D SEOS) is fitted to the force–strain curve to calculate the critical strain and force [10,31]. The detailed derivation of 1D SEOS is given in reference [31], where the authors have studied the uniaxial strain of a layered compound using density functional theory. Moreover, the spinodal equation of state has been successfully applied to investigate the mechanical response of low-dimensional materials [10,19,32,33].

The SEOS can be stated as:(1)$$\sigma =\sigma_{sp}\left(1-{\left(\frac{\epsilon_{sp}-\epsilon }{\epsilon_{sp}}\right)}^{\frac{1}{1-\gamma }}\right)$$

The stress and strain are analytically related, as shown in Equation (1), where $\sigma_{sp}$, $\epsilon_{sp}$ represent spinodal stress and strain, and σ, ϵ represent stress and strain at a particular point, respectively. The fitting parameter *γ* is an exponent whose value depends on the strain’s (stretching or compressing) direction.

At isothermal 0 K, the force can be derived using equilibrium length L, and the internal energy E via the relationship $f=\frac{1}{L}\frac{dE}{d\epsilon }$. Knowing the stress given by Equation (1) and the effective contact area at the interface, the spinodal equation for force can then be written as
(2)$$f=f_{sp}\left(1-{\left(\frac{\epsilon_{sp}-\epsilon }{\epsilon_{sp}}\right)}^{\frac{1}{1-\gamma }}\right)$$
where $f_{sp}$ (spinodal force or critical force) is represented as the maximum force required for the system to break and, hence, indicates the material’s breaking point, referred to as the critical strain (ϵ*_sp_* or ϵ*_c_*).

In this work, we designate critical strain as the transverse strain or separation point. Similarly, critical strength refers to the system’s transverse strength. Furthermore, the proposed state equation can be represented analytically in its energy form. Overall, this appears to be a preliminary step to relate the molecular description of the interface obtained from the first-principles method to the macroscopic mechanical response of the system via the 1D SEOS for the ester monomers interacting with graphene or BN monolayer.

## 3. Results and Discussion

### 3.1. Interaction Energy

Figure 2 illustrates how the molecular-level description of the ester–monolayer complex was obtained. First, the ground-state configurations of fluorinated and non-fluorinated monomers were obtained (Appendix A). Next, the monomer in an in-plane configuration was approached perpendicularly to the surface while keeping the monolayer configuration fixed. It is to be noted that in our previous study of epoxy/BMI monomers interacting with a graphene (or BN) monolayer, a detailed conformational sampling in terms of the orientation of a monomer approaching the surface was investigated [10]. The in-plane orientation was anticipated to be the energetically preferable orientation of the monomer on a surface out of four representative orientations: flip-up, flip-in, vertical, and in-plane. The findings were likewise validated in terms of contact area and atom population at the surface. Compared to the flip-up, flip-in, and vertical orientations, the in-plane orientation has the highest population of atoms and interaction energy [10]. Next, the interaction energy for the representative orientations of fluorinated ester monomer with graphene monolayer was calculated to validate the case for ester monomer. Energy analysis revealed that in-plane orientation is the most favorable orientation (Appendix A). Furthermore, a preference for the longitudinal configuration of monomer over CNT was obtained over the transverse configuration [34].

Following that, the properties such as interaction energy, the effective area of contact, the population of atoms, density of states, and Bader’s charge [32] were calculated to characterize the interface formed by ester monomers with a graphene (or BN) monolayer. The population of atoms is defined as the number of monomer atoms within a distance of 3 Å from the surface. The contact area is the effective area projected by the monomer on the surface of a graphene (or BN) monolayer. By measuring the width and length of the monomer-covered surface, we could compute the effective area of contact. It should be noted that the interplanar distance in the vdW complexes, which includes the monomer–monolayer system under consideration, is reported to be ~3 Å [34]. The interaction energy is defined as **∆***E* = *E_complex_* − *E_monomer_* − *E_monolayer_*, where a negative value of **∆***E* indicates the complex is stable (Figure 3, Table 1).

BN/monomer interactions are stronger than those produced with the graphene due to the polar-π interactions between a polar surface of the BN monolayer and the π system of the monomer [35,36,37]. In contrast, the interaction strength of graphene/ester monomer complexes is governed only by the templating effect [38] of the phenyl groups interacting with a graphitic surface. Note that aromatic rings prefer to align themselves parallel to the surface, forming the π-π stacked configurations [33]. DFT calculations also predict relatively higher stability for the non-fluorinated monomer than the fluorinated monomer (see Figure 3). This effect is associated with the larger area of contact of the non-fluorinated ester (≈102 Å^2^) compared to the fluorinated ester (≈95 Å^2^), leading to an increase in the vdW interactions at the interface. The area of contact for a non-fluorinated ester is relatively large due to the presence of an additional phenyl ring and cyanate group.

This has been affirmed by the population of atoms as well, which is higher for non-fluorinated ester than fluorinated ester monomers (Table 1). Likewise, the calculated values of the area of contact normalized interaction energy show that non-fluorinated ester interacting with graphene has a higher interaction energy per unit area (−0.007 eV/Å^2^) than fluorinated ester (−0.006 eV/Å^2^). Furthermore, the -CF_3_ group of the fluorinated ester induces steric hindrance effects, which may restrict the parallel alignment of the monomer on the surface. This effect has been noted in previous studies of the other fluorinated polymers and CNTs functionalized with fluorine atoms [39,40,41].

To determine whether charge transfer occurs between the constituents in the complex, we now perform Bader’s charge analysis. In both cases, a small charge transfer (<0.1 e) occurred from the graphene (or BN) monolayer to the monomer (Table 1). It then rules out the dominance of the electrostatic interactions at the interface. Furthermore, the electronic density of states (DOS) of the ester complex shows that the DOS of the ester complex is nearly a superposition of the DOS of the individual components (see Appendix A). Moreover, ester monomers do not modify the behavior of the DOS near the Fermi level, suggesting that the interaction at the interface is dominated by the weak vdW forces. Note that Appendix A presents the density of states (DOS) for graphene/BN, and the monomer corresponds to the isolated components, not projected states.

Consequently, the dipole moments of the fluorinated and non-fluorinated ester monomers are predicted as 2.61 Debye and 6.2 Debye, respectively. A relatively high dipole moment of the non-fluorinated monomer, in turn, provides relatively high flexibility in the electronic density, thus facilitating higher vdW interactions at the interface. Ultimately, the results predict that the interface of the ester complexes will be dominated by the noncovalent interactions. However, because of the semi-ionic behavior of the BN, we found a small but remarkable difference in the interaction energies of when compared to graphene complexes: E_interaction_(AroCy F-10_(graphene)_) < E_interaction_(AroCy F-10_(BN)_) and E_interaction_(Primaset PT-30_(graphene)_) < E_interaction_(Primaset PT-30 _(BN)_).

### 3.2. Mechanical Response

The mechanical properties (out-of-plane) of the ester complexes were predicted in terms of critical force (transverse strength), transverse stress, transverse stiffness, and critical strain using the pull-apart setup. In the simulation, an ester monomer was displaced in small steps (≈0.02 Å) normal to the graphene (or BN) surface, starting from its optimized state (equilibrium distance) until ≈3.8 Å of separation (Figure 4, inset). We note that the H atoms of the ester monomers are the nearest-neighbor atoms for the graphene (or BN) monolayer (Appendix A). The critical force calculated during the pull-apart process at various strain points can accurately determine the critical stress and strain.

The point at which the separation occurs is the point of maximum interaction between the monomer and surface and can be called the separation point. Moreover, if further strain is applied to the complex, the force will decrease until it vanishes, indicating that the applied external force has overcome the mechanical strength at the interface of the complex.

Applying the 1D SEOS allows us to derive transverse strength and separation point values for the ester complexes based on the associated fitting parameters σ_sp_, ϵ_sp_, respectively (Appendix A). Furthermore, it is important to note that the pseudocritical exponent (i.e., *γ*) varies between 0.5 and 0.8 (Appendix A) for the ester complexes within the stretching region of the spinodal stress–strain equation of state. For a solid under high pressure, *γ* is reported to be 0.85 [42]. On the other hand, the lower values of *γ* (~0.5) [31] are attributed to the stretching region of the curve.

Table 2 shows the computed transverse stiffness as well as separation point values, which are used to describe the mechanical behavior of the ester complexes. According to our findings, the non-fluorinated ester has a higher transverse strength than the fluorinated ester, with the hierarchical order being AroCy F-10_(graphene)_ < AroCy F-10_(BN)_ < Primaset PT-30_(graphene)_ < Primaset PT-30 _(BN)_. Interestingly, the fluorinated ester is predicted to have a relatively higher separation point than the non-fluorinated ester, enhancing its interfacial load transfer. This fact can be attributed to the influence of fluorine atoms, which cause steric hindrance and interlocking effects in the fluorinated case. Hence, it prevents its separation against the surface, as reported previously [39,40,41].

Figure 5 displays the relationship between the interaction strength and the mechanical response of cyanate ester complexes in terms of transverse stress and stiffness. A stress–strain curve is given in Appendix A, and Appendix A lists the calculated values for which the effective contact area was estimated by utilizing the length and width covered by the ester monomer over the graphene (or BN) monolayer. The results predict that the non-fluorinated ester complexes are stiffer than the fluorinated ester complexes, following the order obtained for the interaction energy values (Table 1). Given that (out-of-plane) stiffness is related to quasi-3D Young’s modulus, a relatively high stiffness value results in a higher Young’s modulus at the interface, one of the polymer composite’s useful features for structural and aerospace applications. It is significant to mention that the calculated separation point values of the cured fluorinated/flCNT and non-fluorinated/flCNT are predicted to be 0.6 and 1.1, respectively, by employing the Polymer Consistent Force Field–Interface Force Field (PCFF-IFF) model in molecular dynamics (MD) computations [43].

## 4. Summary

A state-of-the-art DFT simulation is used to analyze the interfacial properties of fluorinated and non-fluorinated cyanate esters forming complexes with a graphene (or BN) monolayer. We find that the nature of the interface depends on the specific monomer configuration, with the non-fluorinated monomer markedly possessing a higher degree of interaction with the graphene (or BN) monolayer than the fluorinated monomer. This leads to a smaller transverse strength but a higher separation point for the fluorinated ester, which can be explained by the stronger steric hindrance provided by the fluorine groups at the interface. Despite the limited flexibility of the fluorinated ester, which reduces the interaction energy, it appears to provide mechanical interlocking that increases the transverse strain at the interface. Therefore, the present study has enabled us to elucidate the properties at the molecular level that are difficult to determine by experiments. Moreover, we show that the BN-based ester complexes are likely to have a higher mechanical strength than those based on graphene and, thus, can be considered structural materials in aerospace vehicles.

## Figures and Tables

**Figure 1 materials-17-00108-f001:**
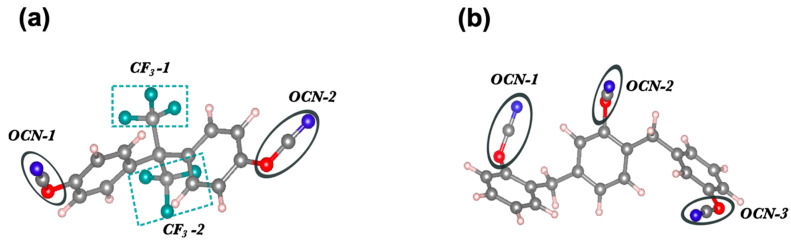
Molecular Representation of (**a**) AroCy-F10 (Fluorinated) and (**b**) Primaset PT-30 (Non-Fluorinated) Cyanate Ester Resin Monomers: Ball and Stick Models with Ester and Fluorine Groups Highlighted. Color codes: C—grey, H—white, N—blue, F—green, O—red.

**Figure 2 materials-17-00108-f002:**
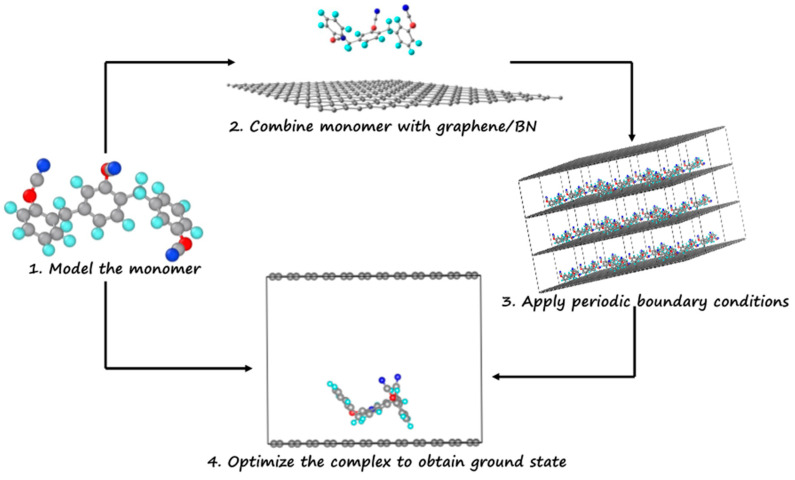
A workflow diagram describing the steps taken to determine the equilibrium structure of the monomer/graphene complex. Color codes: C—grey, H—white, N—blue, F—green, O—red.

**Figure 3 materials-17-00108-f003:**
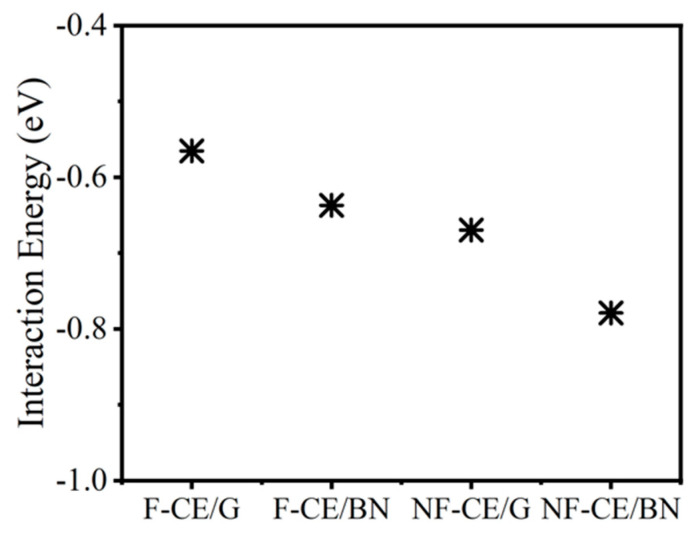
The calculated interaction energy of the ester resins: AroCy-F10 (fluorinated cyanate ester), labeled as F-CE, and Primaset PT-30 (non-fluorinated cyanate ester), labeled as NF-CE. G refers to graphene, and BN refers to a boron nitride monolayer.

**Figure 4 materials-17-00108-f004:**
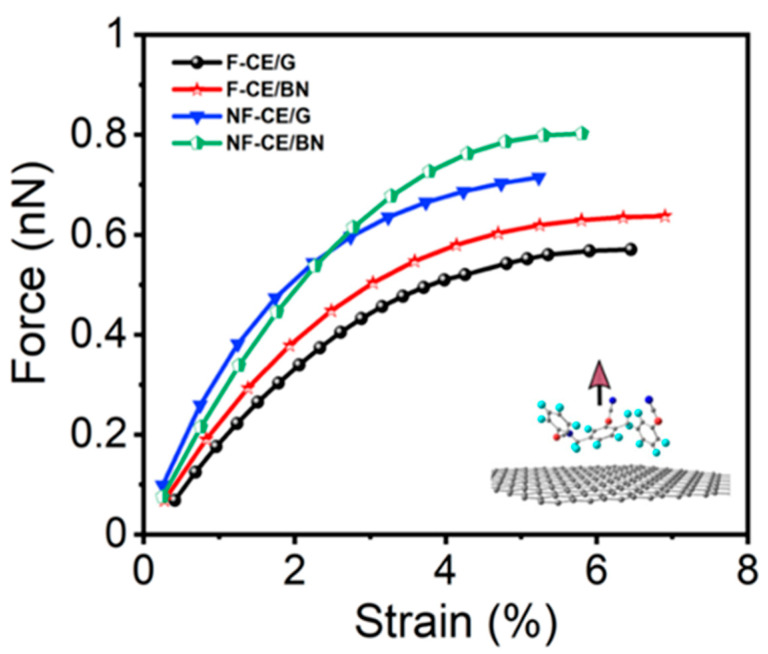
Calculated force vs. transverse strain curve of the ester/monolayer; AroCy-F10 (fluorinated cyanate ester) labeled as F-CE and Primaset PT-30 (non-fluorinated cyanate ester) labeled as NF-CE. G refers to graphene, and BN refers to a BN monolayer.

**Figure 5 materials-17-00108-f005:**
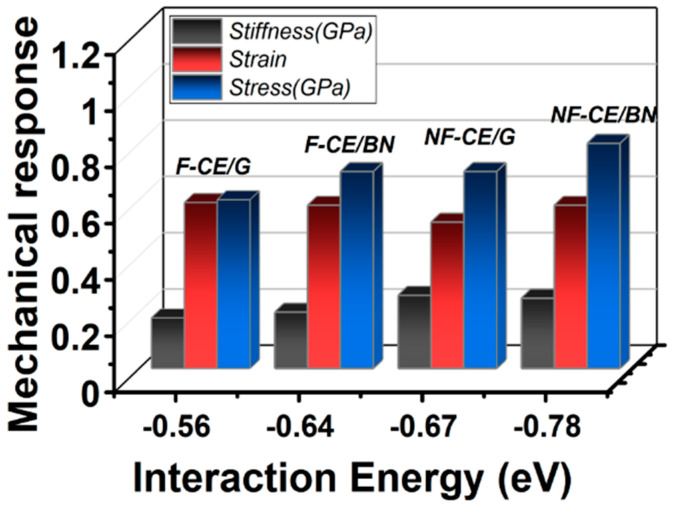
Calculated mechanical response and the interaction energy of the ester complexes; AroCy-F10 (fluorinated cyanate ester) labeled as F-CE and Primaset PT-30 (non-fluorinated cyanate ester) labeled as NF-CE. Also, G refers to graphene, and BN refers to a BN monolayer.

**Table 1 materials-17-00108-t001:** The computed interaction energy (ΔE), the population of atoms, and Bader’s charge (Q) for AroCy-F10 (F-CE) and Primaset PT-30 (NF-CE) interacting with graphene (or BN) monolayer.

Complex		∆E (eV)	Population (%)	Q (e)
Graphene	AroCy-F10 (F-CE)	−0.56	11.4	0.03
Graphene	Primaset PT-30 (NF-CE)	−0.67	13.6	0.04
BN monolayer	AroCy-F10 (F-CE)	−0.64	11.4	0.04
BN monolayer	Primaset PT-30 (NF-CE)	−0.78	13.6	0.04

**Table 2 materials-17-00108-t002:** Predicted (out-of-plane) separation point, and transverse strength of the fluorinated and non-fluorinated cyanate ester complexes formed with graphene (or BN) monolayer.

Complex		(Out-of-Plane) Separation Point ε_c_ (%)	Transverse Strength *f_c_* (nN)
Graphene	AroCy-F10 (fluorinated)	5.9 ± 0.3	0.57 ± 0.01
Graphene	Primaset PT-30 (non-fluorinated)	5.2 ± 0.3	0.71 ± 0.03
BN monolayer	AroCy-F10 (fluorinated)	6.4 ± 0.2	0.63 ± 0.01
BN monolayer	Primaset PT-30 (non-fluorinated)	5.8 ± 0.3	0.80 ± 0.02

## Data Availability

All data generated or analyzed during this study are included in this published article. For more inquiries, contact the corresponding author at gsachdev@mtu.edu.

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
