# Peer review of "A Micromechanical Study of Interactions of Cyanate Ester Monomer with Graphene or Boron Nitride Monolayer"

_materials, 2023, doi:10.3390/ma17010108_

Round 1

Reviewer 1 Report

Comments and Suggestions for Authors

The paper details the computational investigation of mechanical properties of four nanocomposites. The discussion as well as the description of the results is easy to comprehend. The paper is publishable after the clarification of the following points:

Introduction

a) Line 56: FCNTs should be altered to flCNTs.

b) Lines 93 - 96: "It is again noteworthy that we intend to focus on the individual monomer interaction at the reinforcement (monolayer) surface rather than the cured polymer composites in the present study." The authors should elaborate more on that to enrich the discussion. Why focus on the monomer and not the cured polymer? What could be the differences in terms of results?

Methodology section

a) Lines 109 - 110: authors should clarify the meaning of "to ensure 10−5 eV energy"convergence" Did they test the total energy with the cutoff energy? Such small convergence (10−5 eV) is not expected to be reached for total energies, except for total energy/total number of atoms or cohesion energies.

b) Lines 131 - 133: Did the authors also test the "pull-apart" setup by freezing the z-components of the atomic coordinates, and subsequently performing a geometry optimization? Such procedure seems to me more realistic.

Results and Discussion

a) It would be better to state in Fig. S4 that the DOS of graphene/BN and the monomer are not projected states, but DOS computed for the isolated components. This confirmation is important for the discussion at lines 223 - 225, avoiding possible misunderstandings from readers.

b) Figure 5 shows a correlation between the interaction energy and the stress. Do the authors know if there are any data in the literature showing the opposite trend of Figure 5?

Supporting Information

a) Figure S1. There is a problem with the character inside the parentheses in the x-axis.

b) Figures S5 and S6: The description of these figures are not properly placed in the SI. 

Author Response

December 17, 2023

Dear Reviewer,    

            We are thankful for the valuable input provided by the reviewers, which has significantly enhanced the manuscript. Below, we outline the specific revisions made in response to their comments. I have diligently addressed the extensive English revisions as suggested, and the manuscript has undergone thorough revision to meet the recommended standards.

Reviewer 1

The paper details the computational investigation of the mechanical properties of four nanocomposites. The discussion, as well as the description of the results is easy to comprehend. The paper is publishable after the clarification of the following points:

  1. Introduction

1.1 Line 56: FCNTs should be altered to flCNTs.

            Author's Reply: agreed. 'FCNTs' is changed to flCNTs.

1.2 Lines 93 - 96: "It is again noteworthy that we intend to focus on the individual monomer interaction at the reinforcement (monolayer) surface rather than the cured polymer composites in the present study." The authors should elaborate more on that to enrich the discussion. Why focus on the monomer and not the cured polymer? What could be the differences in terms of results?

Author's Reply: Page 4&5: The following (highlighted) text is added per the reviewer's recommendation.

" In this paper, our aim was to understand the atomic-level interactions occurring during the reinforcement process comprehensively. Note that we are aware of the challenges associated with extrapolating results obtained from monomer-level interactions to predict the properties of cured polymer composites. By narrowing our focus to the monomer level, we can elucidate the surface energetics and bonding characteristics that govern the composites's performances. This approach allows us to isolate and analyze the initial stages of the reinforcement process, providing insights into the factors influencing the mechanical properties."

2.0 Methodology section

2.1  Lines 109 - 110: The authors should clarify the meaning of "to ensure 10−5 eV energy convergence." Did they test the total energy with the cutoff energy? Such small convergence (10−5 eV) is not expected to be reached for total energies, except for total energy/total number of atoms or cohesion energies.

Author's Reply:

Page 5 (highlighted text)  The 10-5 eV energy convergence refers to the convergence criteria applied to the total energy/atom during geometry optimization. Specifically, the cutoff of 500 eV for the plane-wave basis set was chosen to achieve this level of convergence. Both parameters were selected based on our prior calculations on such systems [1]. We have modified this point in the revised manuscript for enhanced transparency.

2.2 Lines 131 - 133: Did the authors also test the "pull-apart" setup by freezing the z-components of the atomic coordinates and subsequently performing a geometry optimization? Such procedure seems to me more realistic.

Author's Reply: The 'pull-apart' setup was simulated by moving the monomer away from the surface while keeping the atomic coordinates of the surface frozen. This is based on the assumption that the surface does not undergo relaxation within the given duration of the experiment.

3.0 Results and Discussion

3.1 It would be better to state in Fig. S4 that the DOS of graphene/BN and the monomer are not projected states, but DOS computed for the isolated components. This confirmation is important for the discussion in lines 223 - 225, avoiding possible misunderstandings from readers.

Author's Reply: Page 11 (highlighted text)- Agreed. We have clarified this statement in section 3.1 (results and discussions).

3.2 Figure 5 shows a correlation between the interaction energy and the stress. Do the authors know if there are any data in the literature showing the opposite trend of Figure 5?

Author's Reply: The literature does not currently provide data showing the opposite trend depicted in Figure 5.

4.0 Supporting Information

4.1 Figure S1. There is a problem with the character inside the parentheses in the x-axis.

Author's Reply: We acknowledge the issue with the characters inside the parentheses on the x-axis in Figure S1. We have corrected it in the manuscript's revised version.

4.2 Figures S5 and S6: The description of these figures is not properly placed in the SI. 

Author's Reply: Page 7 (supplementary information)- In the revised manuscript, we have ensured that the descriptions for Figures S5 and S6 are appropriately placed within the Supporting Information (SI) section for better organization and clarity.

References

Sachdeva, G., Lobato, A., Pandey, R., & Odegard, G. M. (2021). Mechanical response of polymer epoxy/BMI composites with graphene and a boron nitride monolayer from first principles. AC

Reviewer 2 Report

Comments and Suggestions for Authors

In this paper, Geeta Sachdeva et al., used micromechanics through the density functional theory methods to analyze interaction energy, charge density, and stiffness of the fluorinated and non-fluorinated cyanate ester monomers composited with graphene and BN. They found that the fluorinated cyanate-ester monomer exhibits lower interaction energy, lower pull-apart force, and higher separation point in comparison to non-fluorinated cyanate ester monomer. I tend to suggest publication after addressing the concerns as follows:

1.     I agree with the authors that polymer composites represent key materials for many applications, however, I think the interaction between polymers and fillers shall be fairly complicated, in most cases, the polymer interaction with filler is quite different from that between monomer and filler, could the authors clarify this crucial point.

2.     Page 4 in Section 3.1, the authors claim that in-plane orientation is favorable, there are two issues shall be clarified:

1)    Have the authors considered the upper layer effect of graphene of BN? I agree with the authors that in-plane orientation shall be energy stable for bottom interaction, how about the influence from the upper layer? How do the authors determine the interlayer distance between two graphene or BN layer?

2)    How do the authors define in-plane periodic boundary? Whether neighboring monolayers influence the in-plane orientation, particularly in the polymer state?

3.     Some other concerns regarding typos like Page 2 line 56 “FCNTs”, and errors of Figure in SI.

Author Response

December 17, 2023

Dear Reviewer,    

            We are thankful for the valuable input provided by the reviewers, which has significantly enhanced the manuscript. Below, we outline the specific revisions made in response to their comments. I have diligently addressed the extensive English revisions as suggested, and the manuscript has undergone thorough revision to meet the recommended standards.

Reviewer 2 Comments and Suggestions for Authors

In this paper, Geeta Sachdeva et al. used micromechanics through the density functional theory methods to analyze interaction energy, charge density, and stiffness of the fluorinated and non-fluorinated cyanate ester monomers composited with graphene and BN. They found that the fluorinated cyanate ester monomer exhibits lower interaction energy, lower pull-apart force, and higher separation point in comparison to non-fluorinated cyanate ester monomer. I tend to suggest publication after addressing the concerns as follows:

2.1 I agree with the authors that polymer composites represent key materials for many applications. However, I think the interaction between polymers and fillers shall be complicated, in most cases, the polymer interaction with filler is quite different from that between monomer and filler, could the authors clarify this crucial point.

Author's Reply: Page 4&5 (highlighted text) -To address this crucial point, we emphasize that our current investigation serves as a first step in understanding the fundamental interactions at the molecular level. The results provide insights into the initial stages of interaction between monomers and fillers, which can influence the subsequent polymerization and composite formation. We acknowledge that transitioning from monomer-filler interactions to polymer-filler interactions involves a more intricate interplay of factors.

In response to the reviewer's comment, we expanded the discussion to clarify our focus on monomer-monolayer interactions. We appreciate the reviewer's valuable input and ensure these considerations are appropriately addressed in the revised manuscript.

2.2.    Page 4 in Section 3.1, the authors claim that in-plane orientation is favorable, there are two issues shall be clarified:

1)    Have the authors considered the upper layer effect of graphene of BN? I agree with the authors that in-plane orientation shall be energy stable for bottom interaction, how about the influence from the upper layer? How do the authors determine the interlayer distance between two graphene or BN layer?

Author's Reply: In the periodic supercell, a vacuum of 20 Å along the z-axis was chosen to prevent interlayer interactions by incorporating.

            Based on findings by Wang et al., [1] it was observed that graphene layers exhibit almost zero binding energy after a 7 Å interlayer distance. Considering this, a distance of 20 Å ensures a significant separation between layers and minimizes any potential interlayer effects. This approach aligns with the reported findings and was implemented to accurately capture the isolated behavior of the graphene or BN monolayer interacting with the ester monomer.

2)    How do the authors define in-plane periodic boundary? Whether neighboring monolayers influence the in-plane orientation, particularly in the polymer state?

      Author's Reply: In our simulations, in-plane periodic boundary conditions were implemented to emulate an infinite and continuous system. This choice aimed to isolate the behavior of individual monolayers and prevent edge effects that might arise in finite systems. While our study primarily focuses on non-polymerized composite systems, we acknowledge that neighboring monolayers may influence in-plane orientation in a polymerized state. We recognize the importance of considering the influence of neighboring monolayers in polymerized systems, and we will incorporate this consideration into outlining potential directions for future studies to explore the impact of neighboring monolayers on in-plane orientation in polymerized systems.

  1. Some other concerns regarding typos like Page 2 line 56 "FCNTs", and errors of Figure in SI.

Author's Reply: Typos, such as the use of 'FCNTs' on page 2, line 56, and errors in figures within the Supplementary Information (SI), have been rectified. The revised manuscript reflects these corrections, addressing the concerns raised by the reviewer.

References

  1. Wang, Z., Selbach, S. M., & Grande, T. (2014). Van der Waals density functional study of the energetics of alkali metal intercalation in graphite. RSC Advances4(8), 4069-4079.